# Robust 3D Pose estimation and Parkinson's Disease classification via Dual-Stage Adaptive Temporal Perception and graph topology modeling network

Min Zuo[1,2], Jialu Li[1], Mingchao Chang[1], Qingchuan Zhang🆔[1]*, Shibo Fan[3]*

**1** National Engineering Research Center for Agriproduct Quality Traceability, Beijing Technology and Business University, Beijing, China, **2** Business School, Beijing Wuzi University, Beijing, China, **3** School of Automation, Wuxi University, Wuxi, Jiangsu, China

* zhangqingchuan@btbu.edu.cn (QZ); ebo19812008@aliyun.com (SF)

## Abstract

This paper proposes a unified skeleton-based framework for 3D human pose estimation and Parkinson's disease classification, integrating a Dual-Stage Adaptive Temporal Perception (DATP) strategy and an Adaptive Graph Topology Modeling Network (AGTM-Net). DATP enhances robustness to joint occlusion and sequence degradation through occlusion-aware interpolation, trend-extrapolated frame padding, and multi-scale spatiotemporal modeling. On the MPI-INF-3DHP dataset with 16 missing joints, DATP achieves 77.72 PCK and 43.57 AUC, outperforming state-of-the-art methods. On Human3.6M, DATP also shows strong generalization with MPJPE reduced to 32.68 mm. For clinical classification, AGTM-Net dynamically models skeletal structure variations and achieves an F1-score of 0.898 and accuracy of 0.881 in distinguishing healthy individuals from Parkinson's patients with a score of 0 based on the "3.9 Arising from Chair" task. Interpretability analyses—based on gradient and perturbation methods—highlight the spine, chest, and hips as decisive joints, aligning with clinical understanding of gait disorders and enhancing the model's transparency and clinical reliability.

## Introduction

Human 3D pose estimation and skeleton-based motion analysis are fundamental tasks with wide applications in healthcare, human-computer interaction, and robotics. In particular, accurate 3D motion modeling from 2D inputs is crucial for clinical scenarios such as Parkinson's disease diagnosis and symptom assessment. However, applying these technologies in real-world clinical settings faces significant hurdles. Clinical videos often suffer from severe joint occlusions (e.g., body parts blocked by furniture or self-occlusion) and varying motion speeds. Standard computer vision models typically struggle to reconstruct coherent skeletons under these conditions,

**Data availability statement:** The code and datasets used in this project are available at the following GitHub repository: https://github.com/JL-Li-st/DATP.

**Funding:** This research was funded by the National Natural Science Foundation of China under Grant No. 62433002 and No. 62476014, the Project of Construction and Support for High-level Innovative Teams of Beijing Municipal Institutions under Grant No. BPHR20220104, and the Beijing Scholars Program under Grant No. 099.The funders had no role in study design, data collection and analysis, decision to publish, or preparation of the manuscript.No authors received a salary from any of the funders.

**Competing interests:** The authors have declared that no competing interests exist.

leading to "jittery" or incomplete data that is unreliable for medical diagnosis. Furthermore, traditional classification models often treat the human skeleton as a static graph, failing to capture the dynamic, subtle motion variations characteristic of early-stage Parkinson's disease (PD).

To address these challenges, we propose a unified skeleton-based framework that integrates robust pose estimation with precise disease classification. First, to ensure data quality, we develop a Dual-Stage Adaptive Temporal Perception (DATP) strategy. This module acts as a robust pre-processor, employing occlusion-aware interpolation to restore missing joints and trend-extrapolated padding to fix incomplete sequences. By using a multi-scale spatial-temporal modeling approach, DATP ensures that the input skeleton data maintains temporal coherence even in the presence of severe occlusion.

Building on this high-fidelity data, we introduce the Adaptive Graph Topology Modeling Network (AGTM-Net) for symptom classification. Unlike static models, AGTM-Net captures the dynamic evolution of skeletal structures during movement via attention-guided updates. For clinical evaluation, this study specifically focuses on the MDS-UPDRS item 3.9 ("Arising from Chair"). This task was selected because it serves as a critical indicator of axial motor impairment and postural instability—symptoms that are often resistant to medication and strongly predictive of fall risk. Based on the features extracted by AGTM-Net, we construct binary classification tasks to distinguish PD patients with varying symptom severity (scores of 0, 1, and 2) from healthy controls.

Finally, to bridge the gap between "black-box" AI and clinical trust, we conduct gradient-based and perturbation-based interpretability analyses. These analyses visualize the key skeletal joints driving the model's decisions, verifying alignment with clinical knowledge of gait disorders.

The most important contribution of this paper is:

(1) A novel DATP framework is proposed, incorporating occlusion-aware interpolation, trend-extrapolated frame padding, unified temporal alignment, and multi-scale adaptive modeling. This innovation significantly improves robustness in scenarios with missing or occluded frames.

(2) An AGTM-Net is designed to dynamically capture evolving joint relationships and extract discriminative features. It is applied to clinically relevant binary classification tasks based on the "3.9 Arising from Chair" score, achieving effective symptom differentiation.

(3) Gradient-based and perturbation-based interpretability analyses are employed to identify critical skeletal joints influencing classification decisions, thereby enhancing model transparency and clinical trustworthiness.

## Related work

### 3D human posture estimation

In recent years, with the popularity of 3D human pose estimation, it has received more and more attention from researchers, and many excellent pose estimation

algorithms have emerged, which have made significant progress in solving the generalization ability, the occlusion problem, and the computational efficiency in different scenarios, etc. Wu et al. [1] proposed a lightweight human pose estimation algorithm based on adaptive feature sensing, which compressed the model volume and at the same time improved the detection efficiency and robustness of the model at the same time. Wang et al. [2] also proposed a new lightweight human posture estimation algorithm based on the non-rigid characteristics of human posture and the diversity of distribution of human landmarks, which significantly improves the accuracy compared with the benchmark algorithm. In order to improve the problem of inaccurate estimation results in human pose estimation due to the complexity of human limbs and environmental factors, Zhang et al. [3] proposed a human pose estimation method based on quadratic generative confrontation, which effectively improves the estimation accuracy of stacked hourglass network (SHN) by generative confrontation training of SHN in two stages. Similarly, to solve the problem of human pose estimation in complex backgrounds, Fu et al. [4] proposed an improved YOLOv7-POSE algorithm, and created a dataset with various shooting angles for training, and the accuracy of the self-constructed dataset was improved by 4% compared with the original YOLOv7 algorithm. Song et al. [5] proposed a hybrid attention adaptive sampling network, which incorporates a dynamic attention module and a pose quality attention module. This network comprehensively takes into account the dynamic information and the quality of pose data. Compared with traditional sampling strategies, such as sparse uniform sampling and keyframe selection based on Convolutional Neural Networks (CNNs), it demonstrates significantly stronger robustness in challenging conditions, including high levels of occlusion, motion blur, and illumination variations. In order to estimate accurate and temporally consistent 3D human motion from videos, Sun et al. [6] proposed the "Bidirectional Temporal Feature for Human Motion Recovery" (BTMR). This model employs bidirectional features, replacing the unidirectional temporal features used in previous studies. Thanks to this improvement, the BTMR model is capable of generating more accurate and temporally coherent 3D human motion. Since the prior template of the human body in the SMPL model is fixed, significant discrepancies may occur in the reconstructed human body shapes when individuals perform vigorous movements such as sports or dancing. To tackle this issue, Wu et al. [7] introduced a parallel-branch network featuring a custom-designed spatiotemporal (ST) branch and an SMPL branch. The 3D joint information from the ST branch is utilized to supervise the 3D joints of the SMPL branch, effectively rectifying the biases in the SMPL model. To address the scarcity of real-world multi-view datasets, Wang et al. [8] introduced the FreeMan dataset, which provides large-scale multi-view image and sequence data, providing a more challenging benchmark for 3D pose estimation. Finally, the PoseIRM method proposed by Cai et al. [9] overcomes the challenges in 3D human pose estimation under unknown camera settings using the Invariant Risk Minimization (IRM) paradigm, enabling the model to adapt to diverse camera configurations with a small number of training samples and improving the generalization ability of the model.

## Classification of disease subtypes

Differentiation of different subtypes of Parkinson's disease is important for treatment and pathologic studies. Currently, Mestre et al. [10] and Dulski at al. [11] researched and found that there is no consensus in the medical field on the classification of different subtypes of Parkinson's disease, mainly due to the extremely heterogeneous nature of Parkinson's disease, the complexity of its etiology and pathogenesis, the lack of reliable biomarkers, and the shortcomings of existing methods. Subtype classification is usually studied in terms of both hypothesis-driven and data-driven subtypes. Hypothesis-driven subtyping relies on predefined clinical criteria, with motor subtypes—specifically Tremor Dominant (TD) and Postural Instability/Gait Difficulty (PIGD)—being the most widely used in clinical practice. However, these classifications often depend on subjective clinical scales (e.g., MDS-UPDRS), which may lack granularity in capturing subtle motor variations. Consequently, data-driven subtyping has gained attention for its objectivity. Deng et al. [12] researched and found that Data-driven subtyping methods do not rely on preconceived assumptions, but rather define the phenotypic characteristics of the disease through the comprehensive analysis of multidimensional data, which has a higher degree of objectivity and data dependence. Parket al. [13] found that Parkinson's disease is heterogeneous in terms of disability and

mortality in a study using follow-up records. Gong et al. [14] analyzed kinesiology data using a machine-learning model, which was characterized by a 79.6% of F1 scores to identify PD kinematic subtypes. Krishnagopal et al. [15] developed a data-driven, network-based trajectory profile clustering (TPC) algorithm for identifying disease subtypes and early prediction of subtypes and disease progression. Fereshtehnejad et al. [16] used a clustering analysis of a comprehensive dataset of baseline (i.e., cross-sectional) data to identify different subgroups by clustering the baseline (i.e., cross sectional) comprehensive dataset, which consists of clinical features, neuroimaging, biospecimens, and genetic information, and then develops criteria for assigning patients to different Parkinson's disease subtypes.Building on these data-driven approaches, this study focuses on utilizing high-fidelity 3D skeletal data to objectively classify motor subtypes related to axial impairment.

### Interpretability analysis

The use of interpretable analysis to explain the results of decision making reveals the parts of the human body that play an important role in decision making, and can effectively improve the credibility of the model, especially in the healthcare industry, which has a high level of risk, and it is important for the model to be able to explain the output. Proper interpretable analysis can also reverse-inspire researchers and promote research progress.Suara et al. [17] study explored the fundamentals of interpretable deep learning and its importance in medical imaging, reviewed various interpretable techniques and their limitations, and focused on the application of Grad-CAM in medical image analysis. The results show that interpretable deep learning and Grad- CAM help to improve the accuracy and interpretability of deep learning models in medical image analysis, and enhance the trust of medical personnel in AI diagnosis. In the perturbation-based approach, Singh et al. [18] researched perturbations are applied to the keypoint information while observing the changes in the model output to determine the keypoints that have a significant impact on the judgment results. Shen et al. [19] In the study of predicting the risk of in-hospital death in patients with chronic heart failure complicated by pulmonary infection using interpretable machine learning, the model was interpreted using the SHAP method. Luo et al. [20] in the study of predicting the risk of in-hospital death in patients with chronic heart failure complicated by pulmonary infection in intensive care, the interpretable deep learning and Grad-CAM can help improve the accuracy and interpretation of deep learning models. The SHAP method was also used in the Interpretable AKI Prediction Study.

## Materials and methods

To address the challenges of severe joint occlusion, incomplete motion sequences, and diverse temporal dynamics in 3D human pose estimation and clinical action recognition, this paper proposes a comprehensive skeleton-based framework. Specifically, we introduce a two-stage Dual-stage Adaptive Temporal Perception (DATP) framework for 3D pose estimation, which includes an occlusion-guided frame padding and trend-based extrapolation mechanism, a unified feature encoding and temporal alignment module, and a multi-scale spatial-temporal modeling strategy with adaptive scale weighting. Furthermore, we propose the Adaptive Graph Topology Modeling Network (AGTM-Net) to extract informative skeletal features. Based on the clinical action score "3.9 Arising from Chair," we construct three binary classification tasks to distinguish Parkinson's patients from healthy individuals [21].

### Datasets

Two public datasets are employed to evaluate the proposed framework across 3D pose estimation and Parkinson's disease classification tasks: the Human3.6M datasets and the REMAP Open datasets.

### Human3.6M datasets

Human3.6M is a large-scale benchmark for 3D pose estimation, containing millions of frames from 11 subjects performing various daily activities. Each frame is annotated with 3D joint positions using a motion capture system. Following standard

protocols, subjects S1, S5, S6, S7, and S8 are used for training, and S9 and S11 for testing. We adopt 2D key points normalized to camera coordinates and predict 17 target joints for evaluation.

### MPI-INF-3DHP datasets

MPI-INF-3DHP is a widely used benchmark for 3D human pose estimation in unconstrained environments. It includes both indoor and outdoor scenes, covering a wide variety of motions, viewpoints, and lighting conditions. The datasets provides synchronized RGB images, 2D key-points, and accurate 3D pose annotations captured using a marker less motion capture system. It is particularly suitable for evaluating the generalization and robustness of 3D pose estimation models under occlusions and real-world conditions.

### REMAP open datasets

The REMAP Open dataset, derived from the PD SENSORS study at the University of Bristol, provides real-world mobility data captured using markerless Microsoft Kinect sensors (640x480 resolution, 30 fps). This study specifically utilizes 403 Sit-to-Stand (STS) episodes, which are expertly annotated according to the MDS-UPDRS item 3.9 criteria, with scores ranging from 0 ('Normal') to 4 ('Unable to arise without help'). Crucially, the dataset records the medication status of participants (defined as 'On' or 'Off' dopaminergic medication), enabling a robust analysis of symptom fluctuations. We selected the STS task as it specifically targets axial motor impairment, a symptom often resistant to medication and a strong predictor of postural instability and fall risk.

The REMAP dataset is publicly available and can be downloaded from: https://github.com/ale152/SitToStandPD.

### Human 3D position estimation based on Dual-Stage adaptive temporal perception modeling

According to Article 32, Clauses 1 and 2 of the Administrative Measures for Ethical Review of Life Science and Medical Research Involving Humans (issued on February 18, 2023), ethical review may be exempted under the following conditions: (1) the research utilizes publicly available data obtained through legitimate means or data derived from the observation of public behavior without intervention; (2) the research is conducted using anonymized information or data. Fig 1 illustrates the overall architecture of the proposed Dual-Stage Adaptive Temporal Perception (DATP) model. Unlike conventional pose estimation methods, DATP addresses challenges such as joint occlusion and temporal modeling difficulties in real-world applications through systematic improvements at both the input enhancement and feature modeling levels. The framework consists of three key components: the occlusion-aware and trend-enhanced input module, the unified feature encoding and temporal alignment module, and the multi-scale perception with adaptive scale weighting module.

### Occlusion sensing and trend enhancement input module

Aiming at the common problems of missing joints and insufficient time boundary information in 2D inputs, this module proposes two new mechanisms based on the traditional occlusion processing and frame supplementation strategies: the local smoothing interpolation mechanism (TLSI); and the frame filling mechanism based on trend extrapolation (TTEP).

### Temporal Locally Smoothed Interpolation (TLSI)

Traditional occlusion complementation mostly uses nearest-neighbor interpolation, which ignores local inter-frame smoothness. For this reason, we propose locally weighted smoothing interpolation (TLSI), which uses the information of neighboring k-frames for weighted complementation.

For the missing key point $(f,j)$, the set of its neighborhood frames is defined as $\mathcal{N}(f)$, and the TLSI interpolation is:

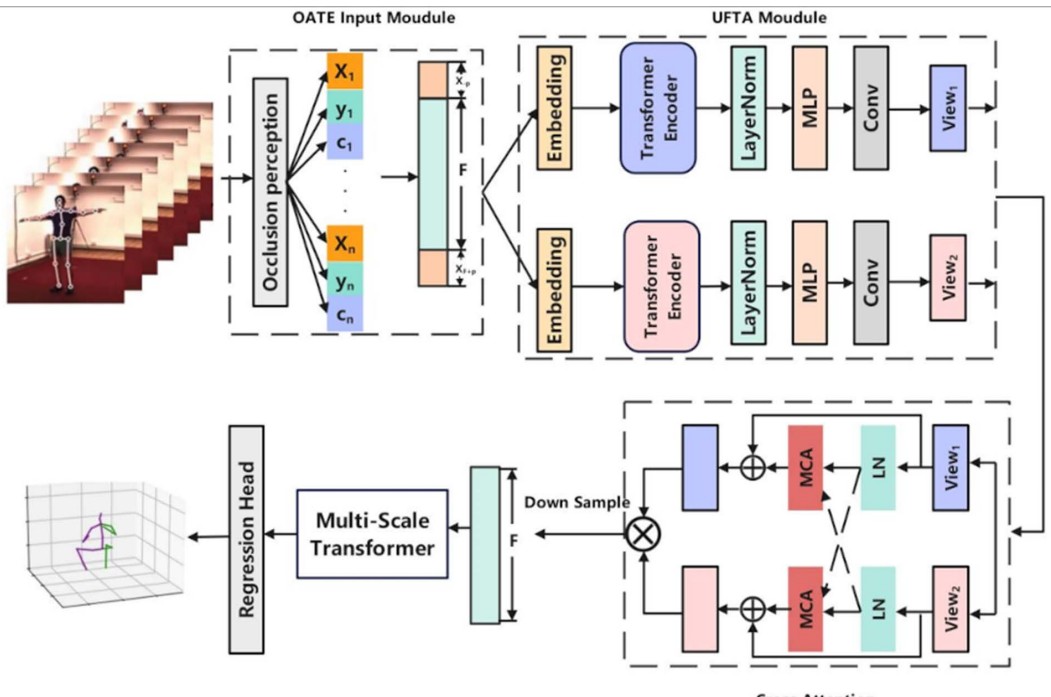

**Fig 1. Overall Architecture of DATP Model.** The DATP model consists of three core components: an OATE Input Module (Occlusion-Aware and Trend-Enhanced Input Module), a UFTA Module (Unified Feature Encoding and Temporal Alignment Module).

$$\mathbf{p}_{f,j} = \frac{\sum_{f' \in \mathcal{N}(f)} w(f, f') \cdot \mathbf{p}_{f',j}}{\sum_{f' \in \mathcal{N}(f)} w(f, f')}$$

(1)

where the weight $w(f, f')$ is the weighting factor between the current interpolated frame $f$ and the neighboring frame $f'$ considering the time distance and confidence level, and $\mathbf{p}_{f,j}$ is the position vector of the $j$-th joint in the neighboring frame $f'$.

$$w(f, f') = \exp\left(-\lambda_t|f - f'|\right) \cdot c_{f',j}$$

(2)

where $\lambda_t$ is the time decay factor and $c_{f,j}$ is the positional confidence of the $j$-th keypoint in the neighboring frame $f'$. The closer the distance and the higher the confidence of the frame, the higher the contribution weight, and the complementary position is guaranteed to be continuous in time and smooth in space.

The final interpolation result is more natural and alleviates the problem of jumps that may be caused by traditional nearest neighbor interpolation.

## Temporal Trend Extrapolated Padding (TTEP)

Conventional frame supplementation by simply replicating the first and last frames can easily lead to repetitive or unnatural information. We propose the trend extrapolation frame-padding (TTEP) strategy to reasonably predict the padding based on the local motion trends of the first and last frames.

Assume that the first $k$ frames of the sequence are $X_{1:k}$, and the last $k$ frames are $X_{F-k+1:F}$, whose linear trend extrapolates forward/backward to generate complementary frames, complementary per $P$ frames.

$$X_{-p} = X_1 + p \cdot \Delta_{\text{head}}, \quad p = P, P-1, \ldots, 1 \tag{3}$$

$$\Delta_{\text{head}} = \frac{1}{k-1} \sum_{i=1}^{k-1} (X_{i+1} - X_i) \tag{4}$$

where $\Delta_{\text{head}}$ is the mean value of the difference of consecutive frames based on the first $k$ frames to infer the motion trend at the beginning of the sequence. Supplemented backward $p$-frames:

$$X_{F+p} = X_F + p \cdot \Delta_{\text{tail}}, \quad \Delta_{\text{tail}} = \frac{1}{k-1} \sum_{i=F-k+1}^{F-1} (X_{i+1} - X_i), \quad p = 1, 2, \ldots, P \tag{5}$$

Where $\Delta_{\text{tail}}$ is the mean value of the difference between successive frames based on the last $k$ frames to infer the motion trend at the end of the sequence, and $X_{F+p}$ is the supplementary backward extrapolated frame.

Linear extrapolation using $\Delta_{\text{head}}$ and $\Delta_{\text{tail}}$ makes the supplementary frames not only continue the original motion trend, but also avoid the static redundancy caused by simple replication. This method makes the supplemental frames conform to the local motion trend, and the boundary features are more natural and continuous, which improves the quality of the two-end time modeling.

## Harmonization of feature coding and time alignment modules

In order to enhance the representation of the input skeleton sequence in the channel and temporal dimensions, this module adopts a two-level feature encoder structure. Each level of this structure contains normalization, feed-forward modeling and residual connectivity, aiming to gradually enhance the interaction between key nodes and construct a unified temporal structure of the frame sequence.

P-frame complementary framing is performed before and after the input to obtain the post-complementary input tensor $X' \in \mathbb{R}^{B \times (F+2P) \times J \times 3}$, which is subsequently rearranged into a temporal feature stream $X''$.

The sequence will be sequentially passed through two residual enhancement modelers to extract temporal dynamic features across nodes. First, the joint representation at the joint channel level is modeled for each frame to obtain the first-stage feature representation:

$$Z_1 = X'' + \text{FFN}_1 \left( \text{LayerNorm}_1(X'') \right) \tag{6}$$

where $Z_1$ denotes the local spatial structure features fused after the first stage of modeling, and $\text{FFN}_1(\cdot)$ is the feed-forward mapping network in the channel direction.

Then, the inter-frame temporal variation patterns are modeled by the second stage to obtain features containing temporal context information:

$$Z_2 = Z_1 + \text{FFN}_2 \left( \text{LayerNorm}_2(Z_1) \right), \quad E_i = \text{Conv1d}(Z_i), \quad i \in \{1, 2\} \tag{7}$$

where $Z_2$ denotes the high-level feature representation incorporating temporal dynamic information. In order to enhance the feature dimension and representation, we map the outputs of the two stages to a unified high-dimensional space separately. $E_1, E_2$ denote the high-dimensional mapping representation of the first and second stage encoding results in the feature space, as the abstract feature flow of the two perspectives.

We introduce the cross-attention mechanism to fuse the two perspectives to enhance the overall modeling capability. After obtaining the encoding of the two perspectives, we introduce a cross-attention mechanism to fuse the two

perspectives to enhance the overall modeling capability of the spatial-temporal structure, and the fusion operation can be expressed as follows:

$$F = \text{CrossAttn}(E_1, E_2) = \text{Softmax}\left(\frac{E_1 E_2^\top}{\sqrt{d}}\right) \cdot E_2 \tag{8}$$

where $E_1$ is used as Query (Q) and $E_2$ is used as Key, Value (K, V), a feature dimension normalization factor. This attention essentially realizes the dynamic weighted aggregation between different views of the feature, so that the fusion feature $F$ can express both local retention and cross-time dependencies. Since the input sequence is extended with $P$ frames before and after; a temporal cropping operation is performed on the fusion feature $F$ to ensure that the length of the output sequence remains the same as the original sequence $F'$.

This operation is essentially down-sampling, removing the complementary frame regions and retaining only the original-length frame features in the middle segments, ensuring that the model output dimensions are consistent with the inputs, thus maintaining structural alignment with subsequent labeling supervision. Ultimately, the encoded unified feature output is represented as $\widetilde{F} \in \mathbb{R}^{B \times F \times C_u}$.

Up to this point, the module has completed the two-stage structural coding of features, multi-view dynamic fusion and sequence length alignment processing, providing structurally unified, dynamically consistent and high-quality feature inputs for subsequent scale modeling and attitude regression.

## Multi-scale perception and scale-adaptive weighting module

In 3D human action sequences, there is significant diversity in the time scales of action changes. For example, fine-grained hand movements usually occur within a short time window, whereas gait changes require a longer temporal receptive field to be modeled. Traditional time series modeling often uses a single scale that cannot simultaneously account for local fine-grained changes and global long-term dynamics.

To this end, we propose a multi-scale perceptual modeling strategy and further introduce the Adaptive Scale Weighting (ASW) mechanism, which enables features at different time scales to automatically adjust their contribution weights according to the dynamic complexity of the input action sequences, thus realizing a more flexible and accurate time-series feature representation.

This module is divided into two main stages: multi-scale spatial-temporal feature extraction; and adaptive weight-based scale fusion.

Fig 2 illustrates the detailed process of the multi-scale transformer encoder.

## Multi-scale spatial-temporal feature extraction

First, the input features $\widetilde{F}$ are subjected to a multi-scale decomposition, and at each scale $d$, a spatial Transformer encoder is first applied to the time series features to extract the dependencies in the local spatial structure:

$$\alpha\widehat{F}_d = F_d + \text{MLP}\left(\text{LayerNorm}\left(F_d + \text{Attention}\left(\text{LayerNorm}(F_d)\right)\right)\right) \tag{9}$$

where $F_d$ is the $d$th scale input feature sequence, and $\widehat{F}_d$ is the output feature sequence after spatial encoding.

In order to separate the dynamic features in different time ranges, the encoded features are scaled in the time dimension as $n_d$, and the length of each segment is $s_d$, which is satisfied $F = n_d \times s_d$, and reshape the features into $\alpha\widehat{F}_d$.

This division can cut long sequences into local segments, which facilitates local dynamic modeling and improves the perception of short-term dynamic changes. The reshape operation splits the original time dimension into two dimensions: (number of subsequences, number of frames per segment), which is ready for subsequent temporal-local Attention modeling.

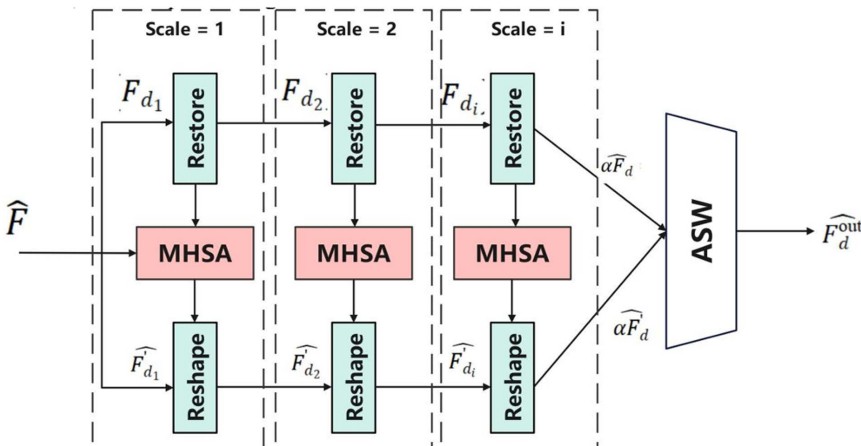

**Fig 2. Structure of the Multi-Scale Perception and Adaptive Scale Weighting Module.** This figure illustrates the architecture of the multi-scale perception and adaptive scale weighting module used in the proposed DATP model.

For each sub-segment interior, the local time Transformer module is applied for modeling:

$$\widetilde{F}_d = \widehat{F}_d + \text{MLP}_T\left(\text{LayerNorm}_T\left(\widehat{F}_d + \text{Attention}_T\left(\text{LayerNorm}_T(\widehat{F}_d)\right)\right)\right) \tag{10}$$

where $\widetilde{F}_d$ is the feature output after temporally localized attention coding. Self-attention is modeled for the time sequences within each local time period. Through the localized attention mechanism, the model can effectively capture short-term temporal change patterns and maintain temporal consistency.

The features of the local segments are re-reduced to a continuous time series form $\widehat{F}_d^{\text{out}}$, where $\widehat{F}_d^{\text{out}}$ is the feature output after the local segment reorganization is reduced to a continuous time series. The operation here ensures that the features are restored to be consistent with the original time length $F$, providing a uniform size for subsequent multi-scale fusion.

## Adaptive Scale Weighting (ASW) Mechanism

Although multi-scale feature extraction can capture dynamic information at different levels, if all scale features are directly fused with equal weights, it may introduce redundant information or weaken the contribution of important scales. Therefore, we further design the scale adaptive weighting mechanism (ASW) to automatically adjust the importance of each scale by learning action sequence features.

(1) Scale Weight Generation.

For each scale $d$, the extracted output features $\widehat{F}_d^{\text{out}}$ are first subjected to global average pooling in the time dimension to compress the scaled feature representation $g_d$ through GlobalAvgPool. The GlobalAvgPool takes the mean on the time axis and extracts the overall profile of the action. Subsequently, $g_d$ is input to a small multi-layer perceptron (MLP) for nonlinear feature transformation to capture scale importance.

Finally, the output is normalized to scale weights in the (0, 1) interval by a Sigmoid activation function; among them, $\alpha_d$ controls the degree of contribution of the $t$-th scale feature in the final fusion, and $\alpha_d$ is sample-adaptive and can be dynamically adjusted according to different action features.

## (2) Multi-scale feature weighted fusion

The features of all scales are weighted and summed according to the corresponding weights to obtain the final multi-scale integrated feature representation:

$$\widetilde{F}_{\text{multi}} = \sum_{d=1}^{D} \alpha_d \cdot \widehat{F}_d^{\text{out}}$$

(11)

Where $D$ is the total number of scales, each scale output $\widehat{F}_d^{\text{out}}$ is dynamically weighted according to the learned weights $\alpha_d$, so that important scale features are strengthened and redundant scales are suppressed. $\widetilde{F}_{\text{multi}}$ is the final fused multi-scale feature representation. This mechanism greatly improves the model's ability to adapt to different temporal dynamics in complex action sequences.

Finally, the fused multi-scale features $\widetilde{F}_{\text{multi}}$ are fed into the regression header, which is mapped to the 3D spatial coordinates of each joint point in each frame by a layer of convolution, and reshaped into the final output format $Y \in \mathbb{R}^{B \times F \times J \times 3}$, which is the 3D joint position predicted for each frame.

### Skeleton-based feature extraction and parkinson's disease classification task

In order to realize the dynamic modeling and feature expression of key skeletal structure relationships in human actions, an Adaptive Graph Topology Modeling Network (AGTM-Net) is proposed in this study.

As shown in Fig 3, the AGTM-Net architecture consists of three main stages: (1) initial graph construction based on the skeletal structure; (2) dynamic adjacency update via attention-guided mechanisms to capture temporal joint interactions; and (3) feature extraction through graph convolutional layers, spatial pooling, and a multi-layer perceptron (MLP) for downstream classification. The attention-based adjacency module enhances the adaptability of skeletal graphs by learning context-dependent joint relations.

This section firstly introduces the overall architecture and key module design of AGTM-Net model, and then builds three independent binary classification tasks based on the extracted skeleton features of "3.9 Arising from Chair" action indicator in Real-world Mobility Activities in Parkinson's Disease (REMAP) datasets. Then, three independent binary classification tasks are constructed based on the extracted skeleton features around the "3.9 Arising from Chair" action indicator in the REMAP dataset.

### AGTM-net skeleton feature extraction model

In order to fully capture the dynamic association relationships between skeletal nodes in a motion sequence, **AGTM-Net** adopts an adaptive modeling strategy based on graph structure. In this model, the skeletal sequence is first encoded as a static topological initial graph structure, and then the adjacency matrix is dynamically updated through the introduction of an attention mechanism to realize the modeling of the changes in the intensity of the joint interactions during the movement. Ultimately, a skeleton feature representation that can effectively distinguish action patterns is generated through multi-layer feature fusion and cross-node feature interaction.

The input skeleton sequence at time step $t$ can be represented as a node feature matrix, i.e., $X_t \in \mathbb{R}^{J \times C}$, where $J$ denotes the number of joints and $C$ denotes the input feature dimension of each node. The initial static skeleton connectivity can be represented as $A_0 \in \mathbb{R}^{J \times J}$ by the adjacency matrix $A_0$, which is defined based on the natural connectivity structure of the skeleton, with 1 indicating the presence of a connection and 0 indicating no connection.

In order to introduce dynamic dependency changes between node pairs, **AGTM-Net** is designed with an adaptive neighbor matrix updating mechanism. For each time step, the dynamic neighbor matrix $A_d$ is calculated as follows:

$$A_d = \text{Softmax}\left(\frac{(X_t W_q)(X_t W_k)^\top}{\sqrt{d_k}}\right)$$

(12)

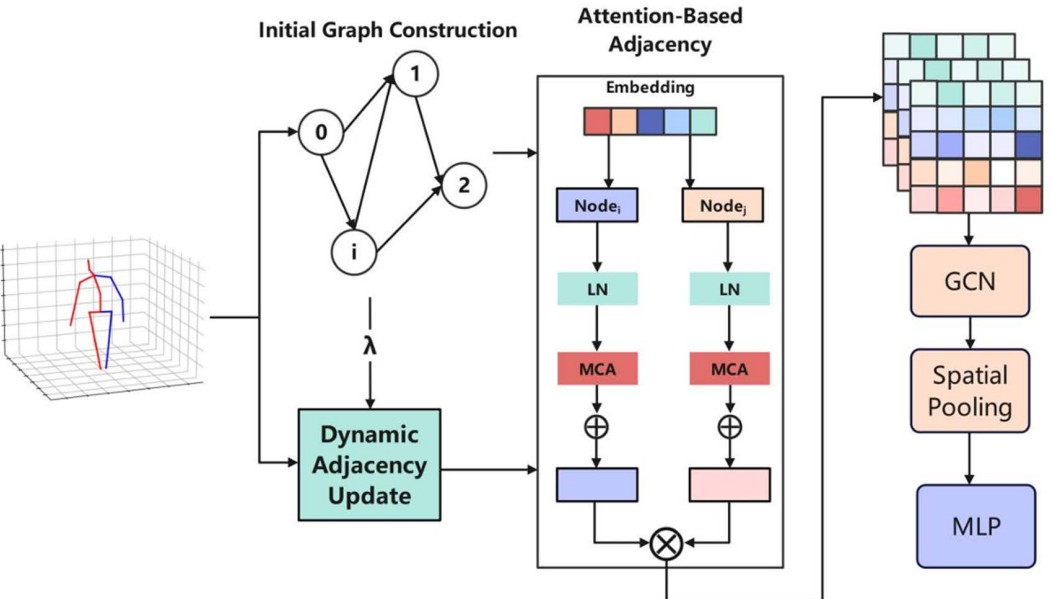

**Fig 3. Architecture of the Adaptive Graph Topology Modeling Network (AGTM-Net).**

where $W_q, W_k \in \mathbb{R}^{C \times d_k}$ is the learnable query and key mapping matrix, and $d_k$ is the dimension of the key vectors used for scaling to stabilize the gradient. The final effective adjacency matrix is defined as a weighted combination of static and dynamic structures:

$$A_{\text{final}} = \lambda A_0 + (1 - \lambda)A_d \tag{13}$$

Where $\lambda \in [0, 1]$ is a hyper-parameter that controls the fusion ratio between static and dynamic structures. After obtaining the updated adjacency matrix, the skeleton features are updated by local information aggregation based on graph convolution as:

$$H_t = \sigma\left(A_{\text{final}}X_t W_f\right) \tag{14}$$

where $W_f \in \mathbb{R}^{C \times C'}$ is the feature transformation matrix, where $C$ is the original feature dimension and $C'$ is the mapped feature dimension, $\sigma(\cdot)$ is the nonlinear activation function, and $H_t \in \mathbb{R}^{J \times C'}$ denotes the feature representation of each joint point after updating at time step $t$.

In order to further model the relationship between different node channels, **AGTM-Net** introduces a Cross Node Feature Interaction (CCI) mechanism, which performs local self-attention enhancement of node features along the channel direction:

$$\widetilde{H}_t = H_t + \text{MLP}\left(\text{LayerNorm}(H_t)\right) \tag{15}$$

where MLP is a two-layer perceptual machine and LayerNorm normalizes the node features. Finally, the skeleton features of all time steps are stacked to obtain the complete skeleton sequence feature representation as

$$H = [\widetilde{H}_1, \widetilde{H}_2, ..., \widetilde{H}_T] \in \mathbb{R}^{T \times J \times C'} \tag{16}$$

where $T$ denotes the total number of time steps in the action sequence. **AGTM-Net** introduces a rich representation of spatial-temporal dynamics changes on the basis of ensuring the rationality of skeleton structure modeling through the above-mentioned neighbor matrix updating, graph convolutional feature aggregation, and channel interaction mechanisms, which provides more discriminative skeleton features for downstream classification tasks.

### Skeleton-based binary classification task design

In order to verify the discrimination ability of the skeleton features proposed by the **AGTM-Net** model in the recognition of Parkinson's disease movement symptoms, the present study is based on the clinical movement score index "3.9 Arising from Chair" in the Real-world Mobility Activities in Parkinson's Disease (REMAP) datasets.

Based on the "3.9 Arising from Chair" clinical movement score index in the Real-world Mobility Activities in Parkinson's Disease (REMAP) datasets, this study constructed two dichotomous classification tasks to discriminate the movement performances of those who scored 0 from the healthy population, those who scored 2 from the healthy population, and those who scored 3 from the healthy population to realize the symptom recognition and *subtype* character modeling.

For each standardized action sequence, the **AGTM-Net** output frame-level skeleton feature sequence is denoted as $H \in \mathbb{R}^{T \times J \times C'}$, where $T$ is the number of time steps, $J$ is the number of skeleton joint points, and $C'$ is the output feature dimension of each node. In order to obtain a uniform temporal representation, all time-step and joint point features are averaged and pooled and compressed to obtain the sample-level global feature vector $g$:

$$g = \frac{1}{T \cdot J} \sum_{t=1}^{T} \sum_{j=1}^{J} H_{t,j} \in \mathbb{R}^{C'}$$

(17)

Next, feature mapping and binary prediction are performed by a two-layer fully connected classification network. The entire binary classification process is performed with each sample pair (i.e., "score = 0 vs healthy population," "score = 1 vs healthy population," and "score = 2 vs healthy population") as an independent task, and the two tasks share the same AGTM-Net feature extractor. The two tasks share the same AGTM-Net feature extractor, but the corresponding classifiers are constructed separately in the training phase for optimization. The final model can be used for Parkinson's disease diagnosis and symptom severity modeling by accurately identifying subtle differences in motor ability.

### Evaluation metrics

To comprehensively evaluate the proposed framework for both 3D pose estimation and skeleton-based classification tasks, multiple evaluation metrics are adopted according to task characteristics.

### Evaluation metrics for 3D pose estimation

Mean Per Joint Position Error (MPJPE) measures the average Euclidean distance between predicted and ground-truth 3D joint locations across all joints and frames. It is defined as:

$$\text{MPJPE} = \frac{1}{N} \sum_{i=1}^{N} \|\hat{p}_i - p_i\|_2$$

(18)

where $N$ denotes the total number of joints, $\hat{p}_i$ and $p_i$ are the predicted and ground-truth positions for the $i$-th joint, respectively.

Percentage of Correct Key-points (PCK) measures the percentage of joints whose prediction errors are within a certain threshold (e.g., 150 mm). It is formulated as:

$$\text{PCK} = \frac{1}{N} \sum_{i=1}^{N} I\left(\|\hat{p}_i - p_i\|_2 < 150\right)$$

(19)

where $I(\cdot)$ is the indicator function that returns 1 if the condition holds and 0 otherwise.

Area Under Curve (AUC) calculates the integral of the PCK curve across varying thresholds, providing a single scalar measure of overall pose estimation performance:

$$\text{AUC} = \int_0^{\tau_{max}} \text{PCK}(\tau)\, d\tau$$

(20)

where $\tau$ represents varying distance thresholds and $\tau_{max}$ is the maximum threshold considered.

### Evaluation metrics for skeleton-based classification

Accuracy measures the proportion of correctly classified samples among all samples. It is defined as:

$$\text{Accuracy} = \frac{TP + TN}{TP + TN + FP + FN}$$

(21)

To further evaluate model performance, especially in unbalanced datasets, precision, recall, and F1-score are reported:

$$\text{Precision} = \frac{TP}{TP + FP}$$

(22)

$$\text{Recall} = \frac{TP}{TP + FN}$$

(23)

$$F1 = 2 \times \frac{\text{Precision} \times \text{Recall}}{\text{Precision} + \text{Recall}}$$

(24)

### Details of implementation

In the 3D pose estimation experiments, the proposed framework incorporates two sequential feature encoding stages, each consisting of 2 encoder layers with 9 attention heads. The input feature dimension is set to dm = 25, and the hidden dimension is set to df = 512. After encoding, a multi-scale temporal refinement module with 5 stages is employed to progressively model motion dynamics across different time scales. All experiments were implemented using the PyTorch framework and trained on two NVIDIA A100 GPUs. The Adam optimizer was used with an initial learning rate of 0.001, and a learning rate decay factor of 0.95 applied after each epoch. During training, horizontal flipping was applied as a data augmentation strategy. The input 2D poses were either obtained from classical 2D pose detectors or directly from ground-truth annotations, depending on the evaluation setting.

For Parkinson's disease movement classification experiments, the number of computational threads was set to 4. The Adam optimizer was adopted with an initial learning rate of 0.0001. The learning rate was decayed by a factor of 0.1 at milestone epochs [30, 40]. Training was conducted for 100 epochs using a single NVIDIA GTX 4060 GPU under the CUDA 11.7 and Python 3.12.4 environment based on the PyTorch framework. During classification, skeleton sequences were normalized, and mini-batch stochastic gradient descent was employed to ensure stable optimization.

## Results

### Experimental results of human 3D pose estimation based on DATP modeling

**Experimental results comparing DATP with state-of-the-art methods on robustness to joint occlusion.** To evaluate the clinical readiness of our model, we benchmarked it against several representative state-of-the-art (SOTA) methods commonly used in computer vision. These baselines serve as proxies for current standard AI capabilities: CNN-based methods (e.g., T3D-CNN) represent traditional deep learning models that are fast but often struggle to capture long-term movement patterns; Transformer-based methods (e.g., MHFormer, STCFormer) represent the current "gold standard" in research, offering high accuracy in ideal conditions by modeling complex temporal relationships. However, these methods are typically not optimized for clinical scenarios where body parts are frequently obscured (occluded). By comparing our DATP model against these established benchmarks, specifically under conditions of "missing joints," we aim to demonstrate the specific advantage of our framework in handling the imperfect video data typical of real-world medical assessments.

To further investigate the robustness of the proposed DATP model under joint-level occlusion, we conducted a comprehensive comparison on the MPI-INF-3DHP datasets. In this evaluation, we progressively increased the number of randomly missing joints per frame and assessed model performance in terms of MPJPE.

As shown in Fig 4, all evaluated methods exhibit increasing MPJPE as the number of missing joints rises. However, the extent of performance degradation varies. Among them, STCFormer displays the steepest error curve, reflecting high sensitivity to partial occlusion. T3D-CNN and MHFormer show moderately rising trends but still suffer from significant accuracy loss under severe occlusion.

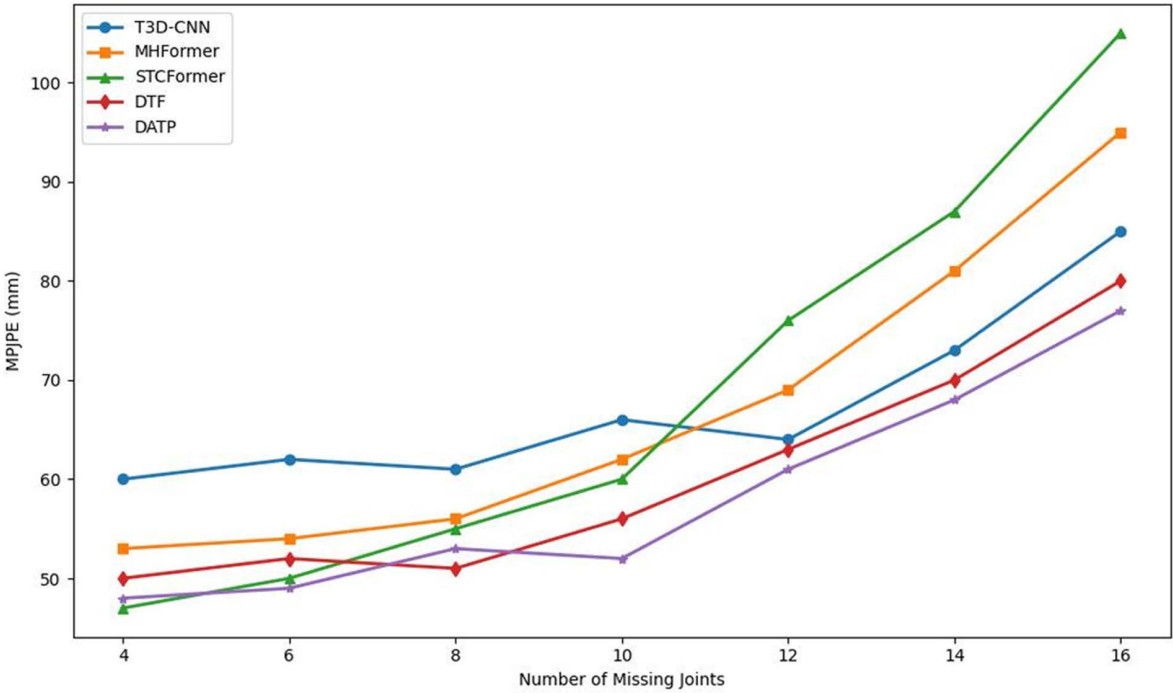

**Fig 4. Comparison of existing SOTA methods.** Including T3D-CNN, MHFormer, STCFormer, and DTF with the proposed fusion model under different levels of missing joints on the MPI-INF-3DHP dataset. As the number of missing joints increases, the proposed method maintains better robustness, exhibiting the lowest MPJPE increase overall.

In contrast, both DTF and the proposed DATP model show relatively stable performance, with DATP achieving consistently lower MPJPE at each occlusion level. This advantage stems from its enhanced temporal alignment and multi-scale feature fusion strategy, which improves the model's ability to capture dynamic spatial-temporal relationships.

Notably, DATP maintains a smooth and moderate MPJPE growth curve, especially under heavy occlusion (e.g., 14 or 16 missing joints), where other models experience a sharp spike in error. This observation highlights DATP's superior generalization ability and resilience under real-world partial observation scenarios. Collectively, the results confirm that DATP not only achieves high accuracy under ideal conditions but also maintains robustness in the presence of substantial joint occlusion.

### Quantitative comparison of DATP and SOTA methods under severe joint occlusion

Table 1 presents a comparative analysis of several state-of-the-art methods (T3D-CNN, P-STMO, MHFormer, STC-Former, and DTF) along with the proposed DATP model on the MPI-INF-3DHP datasets under the condition of 16 missing joints per frame. Two evaluation metrics are considered: Percentage of Correct Key-points (PCK) and Area Under Curve (AUC), which collectively reflect the model's accuracy and robustness.

From the PCK results, it is evident that DATP consistently outperforms DTF across nearly all activity categories. Notably, in high-motion activities such as Exercising, Sitting, and reaching/Crouching, DATP achieves the highest accuracy, surpassing DTF by a significant margin. This suggests that the fusion and temporal alignment strategies of DATP effectively compensate for the structural loss caused by joint-level occlusion.

Similarly, in terms of AUC, the proposed DATP model demonstrates superior generalization ability, consistently achieving higher or second-highest scores across categories. Especially for fine-grained tasks like Sitting and Miscellaneous, DATP shows stronger performance than other SOTA methods, indicating its ability to maintain stability under partial visibility.

On average, DATP surpasses all competing methods in both PCK (77.72) and AUC (43.57), proving its strong robustness and reliability under severe occlusion scenarios.

### Comprehensive evaluation of DATP and SOTA Methods under Human3.6M Protocols

Table 2 and Table 3 presents a detailed comparison of the proposed DATP model with several state-of-the-art (SOTA) methods on the Human3.6M datasets under Protocols 1 and 2, with 16 missing joints per frame using 2D CPN detections. From the average MPJPE results, it can be observed that DATP consistently achieves lower errors than the baseline method DTF under both protocols. Specifically, DATP ranks second among all methods, trailing only behind MHFormer (under Protocol 1) and showing competitive performance across almost all action categories.

**Table 1. Comparison of SOTA Methods with 16 Missing Joints per Frame on MPI-INF-3DHP Datasets Using PCK and AUC Metrics (Input Length = 81).**

| Activity | PCK ↑ | | | | | | AUC ↑ | | | | | |
|---|---|---|---|---|---|---|---|---|---|---|---|---|
| | T3D-CNN | PSTMO | MHFormer | STCFormer | DTF | DATP | T3D-CNN | PSTMO | MHFormer | STCFormer | DTF | DATP |
| Standing / Walking | 29.25 | 46.65 | 55.59 | 35.30 | 83.92 | **85.10** | 9.65 | 21.55 | 28.27 | 15.38 | 51.64 | **52.85** |
| Exercising | 27.50 | 48.94 | 59.48 | 43.06 | 88.03 | **81.12** | 9.93 | 22.21 | 28.94 | 17.78 | 49.36 | **52.21** |
| Sitting | 21.33 | 64.52 | 69.75 | 59.42 | 82.24 | **83.40** | 8.41 | 38.00 | 39.39 | 35.63 | 48.55 | **49.74** |
| Reaching / Crouching | 20.14 | 50.94 | 65.29 | 42.07 | **86.23** | 85.60 | 8.16 | 23.60 | 32.82 | 18.18 | **48.15** | 47.65 |
| On the Floor | 17.32 | 49.27 | 54.26 | 46.39 | 62.33 | **65.88** | 6.01 | 25.89 | 27.55 | 23.90 | 32.80 | **34.76** |
| Sports | 23.73 | 27.76 | 37.45 | 19.99 | 65.87 | **66.92** | 7.35 | 9.76 | 14.64 | 7.06 | **32.17** | 31.84 |
| Miscellaneous | 25.89 | 51.69 | 59.67 | 39.53 | 74.78 | **76.04** | 8.65 | 24.35 | 29.35 | 19.79 | 41.78 | **42.92** |
| Average | 23.59 | 48.54 | 57.36 | 40.82 | 76.49 | **77.72** | 8.31 | 23.62 | 28.71 | 19.68 | 42.72 | **43.57** |

**Table 2. AComparison of the Proposed DATP Model with Existing SOTA Methods in Terms of MPJPE for 16 Missing Joints Per Frame on the Human3.6M Datasets Using 2D CPN (Protocol 1).**

| Method | OAT | Dir. | Disc. | Eat | Greet | Phone | Photo | Pose | Purch. | Sit | SitD | Smoke | Wait | WalkD | Walk | WalkT | Avg. |
|---|---|---|---|---|---|---|---|---|---|---|---|---|---|---|---|---|---|
| PoseFormer-V2 | No | 75.60 | 73.87 | 72.87 | 81.20 | 78.30 | 78.87 | 83.24 | 81.72 | 80.94 | 76.83 | 68.25 | 69.16 | 100.83 | 100.17 | 83.70 | 82.75 |
| STCFormer | No | 71.45 | 70.97 | 64.55 | 79.63 | 74.10 | 69.82 | 81.43 | 82.82 | 78.72 | 74.57 | 67.14 | 66.12 | 120.35 | 125.31 | 84.50 | 82.35 |
| MHFormer | No | 71.62 | 73.38 | 71.29 | 81.96 | 76.20 | 74.59 | 82.64 | 82.12 | 82.01 | 78.84 | 70.64 | 73.02 | 121.21 | 120.15 | 85.10 | 83.65 |
| STE | No | 63.80 | 69.87 | 68.41 | 74.90 | 69.20 | 65.04 | 73.65 | 72.88 | 71.45 | 69.67 | 65.34 | 64.97 | 105.83 | 107.93 | 80.90 | 77.23 |
| PoseFormer | No | 69.30 | 71.92 | 69.11 | 80.40 | 73.70 | 74.89 | 80.52 | 79.33 | 76.52 | 73.18 | 65.55 | 69.92 | 115.87 | 113.85 | 83.30 | 80.47 |
| Ana3D | No | 65.80 | 73.64 | 69.70 | 77.90 | 70.60 | 67.22 | 78.32 | 77.82 | 74.67 | 70.43 | 64.61 | 65.74 | 110.74 | 111.11 | 82.40 | 78.54 |
| VideoPose3D | Yes | 68.17 | 72.65 | 71.35 | 80.30 | 74.20 | 72.28 | 80.43 | 79.63 | 77.26 | 73.69 | 66.71 | 68.42 | 113.15 | 113.21 | 82.70 | 80.61 |
| PoseFormer-V2 | Yes | 76.22 | 74.35 | 73.11 | 84.10 | 80.30 | 78.51 | 86.42 | 85.27 | 83.57 | 79.70 | 71.73 | 72.80 | 101.65 | 104.33 | 87.90 | 84.80 |
| Uplift-Upsample | Yes | 70.79 | 73.92 | 70.50 | 81.40 | 75.60 | 71.94 | 83.40 | 82.90 | 80.66 | 75.44 | 68.35 | 68.98 | 106.33 | 106.49 | 84.10 | 81.24 |
| P-STMO | Yes | 68.04 | 65.05 | 62.84 | 78.20 | 71.40 | 65.43 | 80.41 | 79.55 | 77.45 | 74.68 | 64.34 | 65.53 | 112.43 | 112.96 | 81.60 | 78.35 |
| T3D-CNN | Yes | 55.81 | 54.75 | 49.89 | 68.10 | 61.30 | 55.40 | 67.24 | 67.10 | 63.97 | 60.18 | 52.00 | 52.44 | 87.18 | 87.65 | 72.90 | 67.80 |
| MHFormer | Yes | 68.32 | 66.94 | 65.19 | 79.20 | 72.60 | 69.32 | 81.43 | 81.13 | 78.97 | 75.06 | 67.64 | 67.92 | 109.83 | 108.45 | 83.30 | 79.44 |
| DTF | Yes | 58.40 | 55.24 | 53.13 | 70.50 | 64.10 | 60.21 | 71.34 | 71.15 | 68.09 | 64.86 | 57.82 | 57.43 | 93.00 | 94.27 | 76.20 | 72.85 |
| **Ours (DATP)** | **Yes** | **85.51** | **54.10** | **59.28** | **57.40** | **59.74** | **60.84** | **62.36** | **58.74** | **60.20** | **58.70** | **62.50** | **63.40** | **61.08** | **61.45** | **60.32** | **55.00** |

**Table 3. Comparison of the Proposed DATP Model with Existing SOTA Methods in Terms of MPJPE for 16 Missing Joints Per Frame on the Human3.6M Datasets Using 2D CPN (Protocol 2).**

| Method | OAT | Dir. | Disc. | Eat | Greet | Phone | Photo | Pose | Purch. | Sit | SitD | Smoke | Wait | WalkD | Walk | WalkT | Avg. |
|---|---|---|---|---|---|---|---|---|---|---|---|---|---|---|---|---|---|
| PoseFormer-V2 | No | 51.17 | 52.79 | 51.87 | 51.10 | 60.64 | 52.61 | 57.43 | 67.15 | 61.73 | 57.95 | 55.27 | 55.37 | 92.00 | 89.47 | 63.87 | 60.02 |
| STCFormer | No | 45.01 | 61.65 | 60.55 | 64.64 | 65.17 | 61.75 | 63.89 | 65.19 | 62.32 | 57.47 | 53.56 | 55.06 | 102.87 | 101.13 | 67.59 | 63.17 |
| MHFormer | No | 51.54 | 59.67 | 64.57 | 65.43 | 68.64 | 65.27 | 67.28 | 66.85 | 65.82 | 60.54 | 59.86 | 60.33 | 107.32 | 106.19 | 70.24 | 66.02 |
| STE | No | 43.65 | 54.84 | 60.36 | 59.92 | 61.18 | 58.60 | 60.78 | 61.00 | 59.45 | 55.11 | 53.10 | 52.43 | 92.66 | 90.28 | 65.89 | 61.21 |
| PoseFormer | No | 50.73 | 57.25 | 60.92 | 65.27 | 66.41 | 63.57 | 65.84 | 66.40 | 63.53 | 59.84 | 55.48 | 56.15 | 99.63 | 97.74 | 68.91 | 64.60 |
| Ana3D | No | 51.34 | 50.91 | 54.75 | 60.57 | 61.73 | 57.66 | 60.15 | 61.40 | 58.67 | 54.08 | 50.32 | 51.94 | 95.73 | 93.30 | 65.35 | 60.87 |
| VideoPose3D | Yes | 53.15 | 54.75 | 58.38 | 63.15 | 64.70 | 61.11 | 64.62 | 65.02 | 61.89 | 57.04 | 52.89 | 54.19 | 97.42 | 96.10 | 67.13 | 63.62 |
| PoseFormer-V2 | Yes | 45.34 | 48.52 | 51.65 | 56.84 | 58.21 | 53.67 | 57.32 | 59.04 | 56.52 | 52.30 | 49.26 | 51.17 | 86.28 | 84.99 | 61.45 | 56.37 |
| Uplift-Upsample | Yes | 56.64 | 58.19 | 61.14 | 64.51 | 67.29 | 64.62 | 66.94 | 67.86 | 64.75 | 60.47 | 57.45 | 58.14 | 105.16 | 103.08 | 70.11 | 66.51 |
| P-STMO | Yes | 47.54 | 51.97 | 54.66 | 59.78 | 61.07 | 57.22 | 60.44 | 61.34 | 58.72 | 54.88 | 50.42 | 52.63 | 92.97 | 91.50 | 65.72 | 61.48 |
| T3D-CNN | Yes | 54.72 | 55.48 | 58.19 | 63.10 | 65.82 | 62.17 | 64.40 | 65.35 | 62.78 | 58.64 | 54.21 | 55.89 | 100.11 | 98.43 | 68.43 | 64.67 |
| MHFormer | Yes | 45.42 | 47.21 | 50.87 | 56.47 | 58.04 | 54.13 | 57.38 | 58.62 | 56.19 | 51.81 | 48.14 | 49.30 | 89.27 | 87.53 | 62.34 | 57.14 |
| DTF | Yes | 44.12 | 43.26 | 47.72 | 49.56 | 50.43 | 46.42 | 49.16 | 49.64 | 47.78 | 44.30 | 41.60 | 42.70 | 84.86 | 83.15 | 58.42 | 52.24 |
| **Ours (DATP)** | **Yes** | **43.80** | **43.80** | **41.36** | **43.89** | **43.00** | **41.00** | **43.50** | **43.60** | **41.80** | **41.00** | **43.20** | **43.00** | **42.80** | **43.50** | **43.60** | **43.60** |

Under Protocol 2, which is generally considered more challenging due to the stricter evaluation strategy, DATP still maintains robust estimation capability, outperforming DTF and many other transformer-based approaches in multiple categories such as Eat, Pose, Photo, and Walk Together. This result highlights DATP's generalization ability and robustness against severe joint occlusion scenarios.

In summary, DATP demonstrates superior performance by effectively combining multi-scale temporal-spatial encoding and adaptive temporal padding, which contributes to more stable and accurate pose estimation across varied human actions and occlusion levels.

## Results of the DATP model ablation experiments

To evaluate the contribution of each component in the proposed DATP model, we conducted ablation studies under Protocol 1 and Protocol 2 with 4 random missing joints per frame.

As shown in Table 4, removing both the frame padding strategy and the occlusion confidence leads to a significant performance drop, with MPJPE increasing to 52.12 mm and 41.06 mm, respectively, on the two protocols. When removing only the occlusion confidence, the model still suffers, although slightly less, indicating its standalone contribution to mitigating joint uncertainty.

Similarly, removing only the frame padding strategy increases the MPJPE to 46.15 mm (P1) and 37.02 mm (P2), showing that temporal sequence consistency plays an important role in robust pose recovery. Moreover, excluding the Adaptive Scale Weighting (ASW) module also deteriorates performance, especially under Protocol 2, confirming that dynamic scale weighting is essential for multi-scale feature fusion.

The full model DATP achieves the lowest error (41.32 mm and 32.68 mm), outperforming all ablated variants, demonstrating that each component of the proposed architecture is beneficial and synergistically contributes to the model's robustness under joint occlusion.

## Experimental results of Parkinson's patient type classification based on AGTM-Net

**Experimental results of the classification model based on AGTM-Net.** Table 5 presents the comparative results of the AGTM-Net model against several existing models on the healthy people and patients with a score of 0 datasets. As demonstrated, AGTM-Net achieves outstanding performance across all evaluation metrics, with particularly high scores in precision (0.931) and F1-score (0.898), outperforming all baseline methods. Specifically, AGTM-Net reaches a recall of 0.867, indicating that the model can identify many true positive samples while maintaining strong overall sensitivity. This demonstrates the model's excellent classification performance in practical gait analysis scenarios. Among the competing methods, DeGCN (accuracy 0.820, F1-score 0.859) and SkateFormer (accuracy 0.803, F1-score 0.795) also show competitive results, yet they remain slightly behind AGTM-Net in overall performance. In contrast, CTR-GCN (accuracy 0.765, F1-score 0.762) and HCN (accuracy 0.731, F1-score 0.749) perform noticeably worse in all evaluation aspects,

**Table 4. Ablation Study of Proposed DATP Architecture under Protocols 1 and 2 with 4 Random Missing 2D Joints per Frame.**

| Variant | MPJPE (P1) | MPJPE (P2) |
|---|---|---|
| w/o Frame Padding & Occlusion Confidence | 52.12 | 41.06 |
| w/o Occlusion Confidence | 48.60 | 38.74 |
| w/o Frame Padding | 46.15 | 37.02 |
| w/o Adaptive Scale Weighting (ASW) | 42.85 | 34.05 |
| **Full Model (DATP)** | **41.32** | **32.68** |

**Table 5. Comparative tests between AGTM-Net and other models.**

| Method | Accuracy | Precision | Recall | F1 Score |
|---|---|---|---|---|
| HCN | 0.731 | 0.755 | 0.743 | 0.749 |
| CTR-GCN | 0.765 | 0.768 | 0.757 | 0.762 |
| SkateFormer | 0.803 | 0.810 | 0.790 | 0.795 |
| DeGCN | 0.820 | 0.920 | 0.806 | 0.859 |
| **AGTM-Net** | **0.881** | **0.931** | **0.867** | **0.898** |

suggesting that their capacity to handle complex gait patterns is relatively limited. These results highlight the superior effectiveness of AGTM-Net in gait classification tasks.

**Interpretability analysis of the classification model based on AGTM-Net**

In order to improve the transparency of the model's decision-making, this study conducted an interpretability analysis of the model using two methods: gradient-based interpretability analysis and perturbation-based interpretability analysis. These two methods provide insight into the contribution of each skeletal key point to the final decision outcome, thus helping to understand the model's decision-making rationale when making disease severity judgments.

Taking the classification task between score-0 and healthy individuals as an example, Fig 5 demonstrates the results of the gradient-based interpretability analysis performed on the datasets with a score of 0. As shown in the fig 5, the skeletal key-points that have a large impact on the decision-making results of the model are mainly concentrated in nodes 0, 1, 4, 7, and 8, which correspond to the parts of the human body that are the spine, the thorax, and the hips, respectively. Fig 6 demonstrates the results of the perturbation-based interpretability analysis on the datasets with a score of zero. Through this analysis, it is also found that the key nodes such as spine, chest and hips have a strong influence on the final prediction results These nodes highly overlap with the joint locations in the gradient-based analysis results, indicating that they play a decisive role in making a judgment on the severity of the disease on the datasets with score 0.

By combining gradient-based and perturbation-based interpretable analysis methods, the spine, chest, and hips were found to be the key factors influencing the progression status of Parkinson's disease. This finding not only provides a basis for modeling decisions but also coincides with the clinical physiological understanding of gait disorders, further validating the validity and operationalization of the study model in subtype classification.

## Discussion

This study proposes a unified framework combining DATP for robust pose estimation and AGTM-Net for disease classification. While the results are promising, this study represents a preliminary analysis.

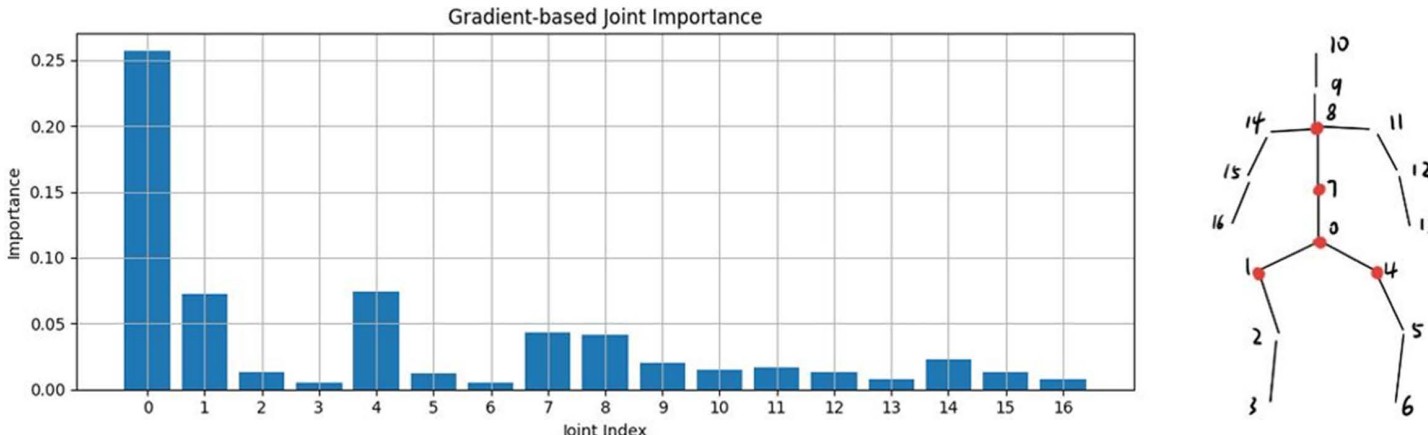

**Fig 5. The gradient-based interpretability analysis performed on the datasets with a score of 0.**

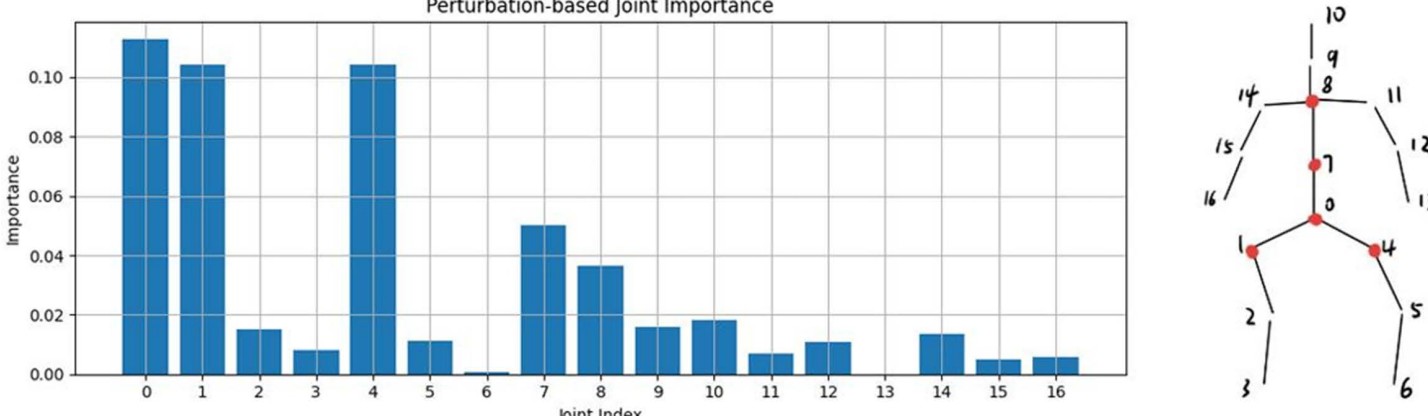

**Fig 6. The perturbation-based interpretability analysis performed on the datasets with a score of 0.**

First, our clinical validation focused exclusively on the MDS-UPDRS item 3.9 ('Arising from Chair'). Although this task is a critical indicator of axial impairment, a comprehensive PD diagnosis requires analyzing a broader range of motor tasks (e.g., gait, finger tapping) to fully capture disease heterogeneity.

Second, the current validation was performed on the REMAP Open dataset. To establish broad clinical utility, larger sample sizes and multi-site validation are necessary to account for geographic variability and diverse patient demographics. Future work will focus on expanding the dataset to include greater variability in PD stages (Hoehn and Yahr stages) to better define disease trajectories and phenotypes.

Finally, while DATP and AGTM-Net are evaluated as separate modules in this paper, they are conceptually unified: DATP reconstructs high-fidelity skeletons from imperfect video data, serving as the necessary pre-processing step for the AGTM-Net classifier.

## Conclusion

The proposed framework in this paper demonstrates excellent performance in 3D pose estimation and Parkinson's disease classification tasks. In the aspect of 3D pose estimation, experiments based on the DATP model show that this model performs outstandingly in handling joint occlusion problems. On the MPI-INF-3DHP datasets, as the number of missing joints increases, the DATP model maintains better robustness compared with other state-of-the-art methods such as T3D-CNN, MHFormer, STCFormer, and DTF, with the smallest increase in MPJPE. Under the condition of severe joint occlusion (16 missing joints per frame), the DATP model outperforms most of its competitors in both PCK and AUC metrics. Under the two protocols of the Human3.6M datasets, it can also achieve a lower MPJPE error, demonstrating good generalization ability and robustness. Ablation experiments further prove that components such as the frame padding strategy, occlusion confidence, and Adaptive Scale Weighting (ASW) module in the model contribute significantly to its performance improvement under joint occlusion.

In the Parkinson's disease classification task, the AGTM-Net model performs excellently. On the datasets of healthy people and patients with a score of 0, its precision reaches 0.931, and the F1-score is 0.898, surpassing all baseline methods. Through gradient-based and perturbation-based interpretability analyses, it is found that the skeletal key points of the spine, chest, and hips have a significant impact on the model's decision-making results. This not only provides a basis for model decision-making but also coincides with the clinical physiological understanding of gait disorders, further validating the effectiveness and operability of the research model in subtype classification. In conclusion, the proposed

framework exhibits strong performance and reliability in both tasks, providing valuable references for research and applications in related fields.

## Supporting information

**S1 Fig. The gradient-based interpretability analysis performed on the datasets with a score of 1.** As shown in the Fig, the skeletal key-points that have a large impact on the decision-making results of the model are mainly concentrated in nodes 0, 1, 4, 7, and 8, which correspond to the parts of the human body that are the spine, the thorax, and the hips, respectively.
(TIF)

**S2 Fig. The perturbation-based interpretability analysis performed on the dataset with a score of 1.** Fig demonstrates the results of the perturbation-based interpretability analysis on the datasets with a score of 1. Through this analysis, it is also found that the key nodes such as spine, chest and hips have a strong influence on the final prediction results These nodes highly overlap with the joint locations in the gradient-based analysis results, indicating that they play a decisive role in making a judgment on the severity of the disease on the datasets with score 1.
(TIF)

**S3 Fig. The gradient-based interpretability analysis performed on the datasets with a score of 2.** Fig shows the results of the gradient-based interpretability analysis on the datasets with a score of 2. As shown in the fig, the key points of the skeleton that have a large influence on the model's decision-making results are mainly concentrated in nodes 0, 1, 4, 7, and 8, which correspond to the parts of the human body that are the spine, the thorax, and the hips, respectively.
(TIF)

**S4 Fig. The perturbation-based interpretability analysis performed on the datasets with a score of 2.** Fig demonstrates the results of the perturbation-based interpretability analysis on the datasets with a score of 2. This analysis reveals that the key nodes, such as the spine, chest, and hips, have a greater impact on the final prediction results. These nodes highly overlapped with the joint locations in the gradient-based analysis results, indicating that they played a decisive role in the judgment of disease severity.
(TIF)

## Author contributions

**Conceptualization:** Min Zuo.

**Investigation:** Mingchao Chang.

**Methodology:** Min Zuo, Jialu Li.

**Software:** Jialu Li.

**Supervision:** Min Zuo, Qingchuan Zhang, Shibo Fan.

**Validation:** Qingchuan Zhang, Shibo Fan.

**Visualization:** Mingchao Chang.

**Writing – original draft:** Jialu Li, Mingchao Chang.

**Writing – review & editing:** Qingchuan Zhang, Shibo Fan.

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
