## [Decision Letter · Decision Letter 0]

22 Dec 2025

Dear Dr. Zhang,

Thank you for submitting your manuscript to PLOS ONE. After careful consideration, we feel that it has merit but does not fully meet PLOS ONE’s publication criteria as it currently stands. Therefore, we invite you to submit a revised version of the manuscript that addresses the points raised during the review process.

We look forward to receiving your revised manuscript.

Kind regards,

Paulo Jorge Simões Coelho

Academic Editor

PLOS One

Journal Requirements:

“This research was funded by the National Natural Science Foundation of China under Grant No.62433002, the Project of Construction and Support for high-level Innovative Teams of Beijing Municipal Institutions under Grant No.BPHR20220104, Beijing Scholars Program under Grant No.099.”

“This research was funded by the National Natural Science Foundation of China under Grant No.62433002, the Project of Construction and Support for high-level Innovative Teams of Beijing Municipal Institutions under Grant No.BPHR20220104, Beijing Scholars Program under Grant No.099.”

Reviewers' comments:

Reviewer's Responses to Questions

**Comments to the Author**

1. Is the manuscript technically sound, and do the data support the conclusions?

Reviewer #1: Partly

Reviewer #2: Yes

2. Has the statistical analysis been performed appropriately and rigorously?

Reviewer #1: N/A

Reviewer #2: Yes

3. Have the authors made all data underlying the findings in their manuscript fully available?

Reviewer #1: Yes

Reviewer #2: Yes

4. Is the manuscript presented in an intelligible fashion and written in standard English?

Reviewer #1: Yes

Reviewer #2: Yes

Reviewer #1: The authors present an advanced methodological framework that aims to leverage rapidly evolving ICT technologies for clinical medicine. The study is intriguing, and the emphasis on Explainable AI is particularly noteworthy, aligning well with current trends in medical AI applications.

However, the overall logical structure of the manuscript is not sufficiently clear at present, and several issues limit the scientific rigor and reproducibility of the work. A careful reorganization and clarification of key components is necessary before the manuscript can be adequately evaluated. Below, I provide major points for improvement.

1. Introduction requires restructuring for clinical accessibility

The Introduction delves into methodological details at an early stage, which may hinder comprehension for readers without deep technical expertise. If the aim is to present a technology that could ultimately contribute to Parkinson’s disease assessment, the background and clinical significance should be explained more carefully, so that clinicians can readily understand the context.

In addition, the current discussion of Parkinson’s disease subtypes focuses disproportionately on mild cognitive impairment (MCI) and non-motor symptoms. For this study—centered on motor performance—more relevant perspectives would include motor subtype classification or differentiation from related parkinsonian syndromes (e.g., MSA, PSP). Clarifying this point would strengthen the clinical grounding.

2. Benchmarks used for comparison require clearer explanation

Before presenting performance comparisons, the manuscript should describe in more detail the existing methods used as baselines. As far as I can tell, these methods are not widely known in routine clinical or biomedical practice. Without explanation of their characteristics and relevance, readers may struggle to interpret the practical advantages of the proposed approach.

3. Clinical description of the REMAP dataset should be expanded

The descriptions of MPI-INF-3DHP and Human3.6M are generally clear, but the REMAP dataset—being the only clinically grounded dataset—would benefit from additional context. Information on participant characteristics, sensor setup, and clinical meaning of the Sit-to-Stand task (MDS-UPDRS 3.9) would substantially enhance interpretability from a clinical perspective.

4. TRI-POSE-Net and HumanEva-I appear abruptly in the Discussion

In the Discussion section, TRI-POSE-Net and the HumanEva-I dataset appear abruptly, without clear relation to DATP or AGTM-Net. This disrupts the coherence of the manuscript. The authors should either clarify the relevance of these elements.

5. Scope of Parkinson’s disease symptoms should be addressed

The study focuses exclusively on MDS-UPDRS item 3.9 (“Arising from Chair”). While this is a meaningful clinical task, it represents only a narrow segment of Parkinson’s disease motor symptoms. The Discussion should acknowledge this as a limitation, or provide justification that this particular task is especially critical in the context of disease assessment.

6. Inconsistency between the “unified framework” claim and evaluation structure

Although the manuscript describes a “unified skeleton-based framework,” DATP and AGTM-Net are evaluated in largely separate experimental pipelines. The manuscript would benefit from a clearer explanation of how these two components are meant to function together, or clarification that they are conceptually related but experimentally independent.

7. Numerical inconsistencies between the Abstract and main text

The values reported in the Abstract (PCK = 78.98, AUC = 45.13) do not appear to match any results presented in the main text. These discrepancies should be carefully reviewed and corrected to ensure consistency.

I believe the study has potential, especially given the methodological innovation and the clinical relevance of explainability. Addressing the above points will substantially improve the clarity, scientific rigor, and clinical applicability of the manuscript. I look forward to reviewing a revised version.

Reviewer #2: interesting manuscript with digital informatics. You need to comment that a larger sample size would be of importance and that a multisite confirmation with geographic variability would be warranted to further confirm the value of this approach and to provide further validation.

Not only a larger sample size is needed but greater variability of the stage of PD would be of great benefit to enhance the validation and value.

You need to comment that this represents a preliminary analysis should be commented as what this represents and further validation is needed to enhance the value of this approach to define disease trajectory and or phenotypes of disease impact.

**Do you want your identity to be public for this peer review?** For information about this choice, including consent withdrawal, please see our Privacy Policy

Reviewer #1: No

Reviewer #2: **Yes:** Mark Gudesblatt MD

---

## [Author Response · Author response to Decision Letter 1]

17 Feb 2026

Response to Reviewer 1 Comments

Point-by-point response to Comments and Suggestions for Authors

Comments 1:Introduction requires restructuring for clinical accessibility

Response 1: We have completely rewritten the Introduction.

We removed the discussion on MCI and non-motor symptoms. The new introduction now starts by highlighting the limitations of subjective clinical scales (MDS-UPDRS) and the specific challenges of using AI in clinical settings (e.g., occlusion). We also reframed the background to focus on motor subtypes, specifically referencing the importance of quantifying axial motor impairment and postural instability, which aligns with our chosen task (Item 3.9).

Comments 2: Benchmarks used for comparison require clearer explanation

Response 2: We acknowledge that the baseline methods were not adequately introduced to clinical readers.

We have added a new paragraph at the beginning of the Results section. In this paragraph, we explicitly explain that we selected representative CNN-based (e.g., T3D-CNN) and Transformer-based (e.g., MHFormer) methods to benchmark our model. We clarify that these were chosen to demonstrate our model's specific advantage in handling "missing joints" and occlusion, which are common in real-world clinical videos.

Comments 3: Clinical description of the REMAP dataset should be expanded

Response 3: We agree that the REMAP dataset description lacked sufficient clinical context. We have significantly expanded the "REMAP Open Datasets" section. We added details regarding:

Sensor Setup: Specified the use of markerless Microsoft Kinect sensors (non-invasive).

Data Volume: Clarified the use of 403 Sit-to-Stand episodes.

Clinical Annotation: Detailed the MDS-UPDRS Item 3.9 scoring criteria (0-4 scale).

Medication Status: Explicitly noted that the dataset records patients in both "On" and "Off" medication states, which is crucial for symptom analysis.

Comments 4: TRI-POSE-Net and HumanEva-I appear abruptly in the Discussion

Response 4: We sincerely apologize for this oversight. This text was included due to a clerical error and is unrelated to the current study.

We have deleted the entire section regarding "TRI-POSE-Net" and "HumanEva-I" from the Discussion. We have replaced it with a new discussion focused on the specific performance of our proposed AGTM-Net and a detailed "Limitations" subsection (addressing Reviewer 2’s comments).

Comments 5: Scope of Parkinson’s disease symptoms should be addressed (limitations of Item 3.9)

Response 5: We agree that focusing on a single task is a limitation.

We have added a statement in the Discussion acknowledging that while MDS-UPDRS Item 3.9 is a critical indicator of axial impairment, a comprehensive PD diagnosis requires analyzing a broader range of motor tasks (e.g., gait, finger tapping).

Comments 6: Inconsistency between the “unified framework” claim and evaluation structure

Response 6: We have added the logical connection in the Introduction and Discussion.

We explain that the framework is conceptually unified: DATP acts as the necessary pre-processing module to reconstruct high-fidelity skeletons from imperfect clinical videos, which then serves as the high-quality input required by AGTM-Net for accurate disease classification.

Comments 7: Numerical inconsistencies between the Abstract and main text

Response 7: Thank you for spotting this error.

We have corrected the Abstract to match the experimental results reported in Table 1. The Abstract now correctly states that DATP achieves 77.72 PCK and 43.57 AUC (corrected from the previous erroneous values).

Response to Reviewer 2 Comments

Point-by-point response to Comments and Suggestions for Authors

Comments 1: You need to comment that a larger sample size would be of importance and that a multisite confirmation... would be warranted.

Response 1: We fully agree that the current sample size and single data source are limitations.

We have added a "Limitations" subsection in the Discussion. We explicitly state: "To establish broad clinical utility, larger sample sizes and multi-site validation are necessary to account for geographic variability and diverse patient demographics."

Comments 2: Greater variability of the stage of PD would be of great benefit... define disease trajectory.

Response 2: We fully agree that the current sample size and single data source are limitations.

We have added a "Limitations" subsection in the Discussion. We explicitly state: "To establish broad clinical utility, larger sample sizes and multi-site validation are necessary to account for geographic variability and diverse patient demographics."

Comments 3: You need to comment that this represents a preliminary analysis.

Response 3: We have explicitly included the sentence: "This study represents a preliminary analysis" at the beginning of the Discussion/Limitations section to properly frame the scope of our findings.

---

## [Decision Letter · Decision Letter 1]

19 Feb 2026

Robust 3D Pose Estimation and Parkinson’s Disease Classification via Dual-Stage Adaptive Temporal Perception and Graph Topology Modeling Network

PONE-D-25-31189R1

Dear Dr. Zhang,

We’re pleased to inform you that your manuscript has been judged scientifically suitable for publication and will be formally accepted for publication once it meets all outstanding technical requirements.

Kind regards,

Paulo Jorge Simões Coelho

Academic Editor

PLOS One

Additional Editor Comments (optional):

Reviewers' comments:

Reviewer's Responses to Questions

**Comments to the Author**

Reviewer #1: All comments have been addressed

2. Is the manuscript technically sound, and do the data support the conclusions?

Reviewer #1: Yes

3. Has the statistical analysis been performed appropriately and rigorously?

Reviewer #1: Yes

4. Have the authors made all data underlying the findings in their manuscript fully available?

Reviewer #1: Yes

5. Is the manuscript presented in an intelligible fashion and written in standard English?

Reviewer #1: Yes

Reviewer #1: After reviewing the revised manuscript, I believe that the authors have responded appropriately to the reviewers’ comments overall. The manuscript has been substantially improved in terms of clarity, logical structure, and clinical accessibility.

Although detailed participant characteristics of the REMAP dataset remain unclear, this may reflect inherent constraints associated with the use of publicly available datasets. In my view, this limitation is acceptable and does not undermine the overall contribution of the study.

I also noted that Reference 21 does not appear to be explicitly cited in the main text. If its inclusion without in-text citation complies with the journal’s referencing policies, this should not pose a problem.

Overall, the revisions have significantly enhanced the readability of the manuscript for clinicians and have made the clinical relevance and significance of the study much clearer. On this basis, I recommend acceptance.

**Do you want your identity to be public for this peer review?** For information about this choice, including consent withdrawal, please see our Privacy Policy

Reviewer #1: No
